# Strain and Substrate-Induced Electronic Properties of Novel Mixed Anion-Based 2D ScHX_2_ (X = I/Br) Semiconductors

**DOI:** 10.3390/nano14171390

**Published:** 2024-08-26

**Authors:** Ashima Rawat, Ravindra Pandey

**Affiliations:** Department of Physics, Michigan Technological University, Houghton, MI 49931, USA; pandey@mtu.edu

**Keywords:** CBM, VBM, heterostructure, band gap

## Abstract

Exploration of compounds featuring multiple anions beyond the single-oxide ion, such as oxyhalides and oxyhydrides, offers an avenue for developing materials with the prospect of novel functionality. In this paper, we present the results for a mixed anion layered material, ScHX_2_ (X: Br, I) based on density functional theory. The result predicted the ScHX_2_ (X: Br, I) monolayers to be stable and semiconducting. Notably, the electronic and mechanical properties of the ScHX_2_ monolayers are comparable to well-established 2D materials like graphene and MoS_2_, rendering them highly suitable for electronic devices. Additionally, these monolayers exhibit an ability to adjust their band gaps and band edges in response to strain and substrate engineering, thereby influencing their photocatalytic applications.

## 1. Introduction

Mixed-anion materials and their 2D counterparts containing more than one anionic species in a single phase have recently attracted significant attention due to their potential to offer novel and attractive functionalities not observed in conventional “single-anion” materials [1]. These materials present exciting opportunities for the development of new properties and applications, expanding the horizons of materials science and technology. By incorporating multiple anionic species, mixed-anion compounds can exhibit unique electronic, structural, and chemical characteristics, paving the way for advancements in energy storage, catalysis, optoelectronics, and other cutting-edge fields. They are synthesized by combining various ionic species, and their appeal lies in the ability to harness disparities in anionic characteristics. These characteristics include ionic radii, charge, electronegativity, and polarizability. By manipulating these anionic attributes, new avenues have been opened for diverse and promising applications of mixed anion compounds. They extend to fields such as visible-light photocatalysts [2,3,4], ion conductors [5,6], thermoelectrics [7], and superconductors [8]. For instance, these compounds exhibit unique properties that make them highly efficient in harnessing sunlight for various chemical processes, including water splitting and environmental pollutant degradation. In the context of ion conductors, mixed-anion compounds can facilitate the controlled movement of ions, enabling their use in solid-state batteries and fuel cells [1]. Additionally, in the quest for superconductors, these compounds present intriguing possibilities for achieving higher-temperature superconductivity, which could revolutionize energy transmission and storage. The unique properties of mixed-anion compounds, such as their tunable electronic structures and enhanced chemical stabilities, make them promising candidates for the development of next-generation superconductors operating at higher temperatures.

In essence, mixed-anion compounds represent a captivating avenue of exploration in materials chemistry, offering a rich tapestry of opportunities for developing novel materials with exceptional properties. These compounds hold the potential for diverse applications across multiple scientific and technological domains, including energy storage and transmission, catalysis, electronics, and photonics. The continued study and development of mixed-anion compounds could lead to significant advancements in these fields, paving the way for innovative solutions to some of the most pressing challenges in science and technology today.The scientific research on mixed-anion compounds has primarily been focused on oxide-based compounds, including oxynitrides and oxyfluorides [1]. More recently, oxygen-free mixed-anion compounds like Li_6_PS_5_Br [9], Li_10_GeP_2_S_12_ [10], CrSBr [11], and Cs_2_PbI_2_Cl_2_ [12], have gathered a lot of attention by experimentalists. One significant advantage of these oxygen-free compounds is that the lack of oxygen mitigates electrostatic interactions with conducting ions and reduces migration barriers within the materials.

First principle calculations have become increasingly important in the quest for new 2D materials with novel functionalities and tailored properties. High-throughput computational methods have significantly driven the rapid advancement in this field. Using high-throughput screening on 5619 experimentally known layered 3D compounds, Mounet et al. identified 1036 candidates that could potentially be exfoliated into monolayer 2D materials [13]. Following this pioneering work, numerous other 2D materials have been discovered using the lattice decoration of existing prototypes of 2D materials, further enriching the landscape of 2D materials research and broadening the potential for innovative applications across various technological domains [14,15,16].

One simpler mixed-anion compound, LaHBr_2_, was successfully synthesized in 1992 [17], and has also been explored theoretically [18]. By examining the layered structure of LaHBr_2_, we aimed to identify and investigate similar stable 2D counterparts that can be experimentally synthesized. In this context, we propose a new 2D ScHX_2_ (X = Br/I) monolayer investigating its structural, mechanical, and electronic properties. The stability of this novel 2D material will be confirmed by employing the density functional theory (DFT) following the three major criteria: lattice dynamical stability, thermodynamic stability, and mechanical stability. This study also involves strain-induced variation in the electronic properties as well as substrate-supported aspects of these mixed anion-based monolayers. Many 2D materials have also been discovered through computational screening based on experimental bulk compounds.

## 2. Computational Details

The Vienna ab initio simulation package (VASP) was used to perform DFT calculations, based on the projected augmented wave (PAW) pseudo-potential [19,20,21,22]. The Perdew-Burke-Ernzerhof form of the generalized gradient approximation (PBE-GGA) [23] has been embraced to describe ion–electron interaction with an energy cut-off set to 500 eV along with the van der Waals (vdW) D3-correction term proposed by Grimme [24]. The convergence criteria for energy and the Hellman-Feynman force acting on each atom were set to 10^−6^ eV and 0.01 eV/Å, respectively. The Brillouin zone was sampled using a Γ centered k-point grid of size (12 × 12 × 1), and periodic image interactions were minimized by using a vacuum of 20 Å along the z-axis direction. The density functional perturbation theory (DFPT) has been used to calculate the phonon frequencies using the PHONOPY package [25,26,27] in a periodic supercell of 2 × 2 × 1. It is to be noted that the 2D structure of ScHX_2_ monolayers was derived from the bulk analog of LaHBr_2_.

## 3. Results and Discussion

### 3.1. Ground State Configurations

ScHX_2_ monolayers belong to the hexagonal symmetry having a space group P6m2 as shown in Figure 1a,c. The simulation box comprises of a hexagonal unit cell with lattice constants a = b and c > 20 Å, and the α = β = 90°,γ = 120°. The lattice consists of 1-Sc, 1-H, and 2-X(Br/I) atoms. The calculated lattice constants and bond lengths are listed in Table 1. The band structures in Figure 1b,d show indirect band gaps of 2.98 eV for ScHBr_2_ and 2.49 eV for ScHI_2_.

For a better understanding of the nature of bands, we have calculated the atom-decomposed band structures. From Figure 1b,d, we can understand that the valance band maxima (VBM) is dominated by the Br/I atoms with a smaller contribution from H atoms, and the conduction band minima (CBM) is composed mainly of Sc atoms. This has also been confirmed using the band decomposed charge density at the VBM and CBM as depicted in Figure 2. In addition, we have also provided the partial density of states in Figure A1 to observe the above contribution in the Appendix A. The calculated ionization potential (IP), work function (ϕ), and electron affinity (EA) have been listed in Table 1. Note that the IP, ϕ, and EA are crucial properties of semiconductors, especially in the context of their applications in various devices and systems, such as metal-semiconductor junctions, photovoltaic cells, and environmental catalysts. We find that the values of the ϕ, EA, and IP for both monolayers are comparable to the existing 2D materials like graphene, MoS_2_, WS_2_, phosphorene, etc. [28,29,30]. In addition the bandgaps are comparable to 2D- NiPS_3_, γ-CuBr, GeSe_2_ [31] and higher than that of some of the ferromagnetic 2D materials containing more than two elements (Mn_2_FeC_6_N_6_ with E_g_ = 1.83 eV, NiRe_2_O_8_ with E_g_ = 1.58 eV) [32].

The effect of spin-orbit coupling (SOC) in both valence and conduction bands is displayed in Figure A2. With the incorporation of SOC, the degeneracy in bands appears though the bandgap remains nearly the same. Hence the SOC effect is not considered in further calculations. Also since the GGA-PBE underestimates the bandgap we have calculated the HSE06 band structures which show a bandgap of 4.02 eV and 3.38 eV for ScHBr_2_ and ScHI_2_ monolayer and provided them in Figure A3.

### 3.2. Stability

In assessing the stability of the ScHX_2_ monolayers, the first step involves evaluating their thermodynamic stability by calculating cohesive energies (*E*_*cohesive*_) using the following formula [33]:(1)$$E_{cohesive}=\frac{E_{ScHX2}-E_{Sc}-E_{H}-E_{X}}{N}$$
where $E_{ScHX2}$, is the total energy of the monolayer, $E_{Sc}$ is the total energy of a single *Sc* atom, $E_{H}$ is the total energy of a single *H* atom, $E_{X}$ is the total energy of the *X*: Br/I atoms, and *N* is the number of atoms in the unit cell.

The calculated cohesive energies of ScHBr_2_ and ScHI_2_ monolayers are −3.43 and −3.10 eV/atom, respectively. By comparing the calculated cohesive energies with those of the synthesized 2D materials such as phosphorene (−3.44 eV/atom) [34], silicene (−3.94 eV/atom) [35], Be_2_C monolayer (−4.84 eV/atom) [36], it is evident that the ScHX_2_ monolayer (X: Br/I) demonstrates robust structural stability.

Next, to analyze the lattice dynamic stability, we have performed calculations for the phonon band dispersion using Density Functional Perturbation Theory (DFPT), as illustrated in Figure 3. The absence of negative modes in the phonon branches for both monolayers across the entire Brillouin zone confirms the dynamic stability of both monolayers. Due to the increase in the atomic mass from Br to I, the optical phonon branches shift downwards by 5 THz, as expected (Figure 3).

The Born Huang stability criteria have been taken into consideration to investigate the mechanical stability [37].

It includes conditions for a hexagonal system as: $C_{11}>\left|C_{12}\right|$, $C_{22}>0$, $C_{66}>0$, and $C_{11}C_{22}-C_{12}^{2}>0$. Using the finite difference method [38], the calculated elastic constants, which are listed in Table 2, satisfy the Born Huang Stability criteria. In addition, Ab-initio molecular dynamic simulations (AIMD) by using the canonical NVT ensemble have been carried out to verify the thermal stability at 300 K. The simulations were carried out for 10 ps with a time step of 1 fs using a 4 × 4 periodic supercell. As shown in Figure A4, the total energy fluctuation within the simulation time is very negligible, thereby confirming the thermal stability of the ScHBr_2_ monolayer.

### 3.3. Mechanical Properties

Young’s modulus is a measure of a material’s stiffness. For materials with a higher Young’s modulus, it means that they require a greater amount of stress to induce a given amount of deformation or strain in their structure. In simple terms, they are less prone to deformation and can withstand more stress before changing shape significantly. This property is desirable in applications where structural integrity and resistance to deformation are crucial. Conversely, materials with a lower Young’s modulus are more flexible and deform more readily under stress. This characteristic can be advantageous when applying strain to induce electric polarization. In such cases, materials with lower Young’s modulus are preferred because they respond more readily to strain-induced changes and can exhibit tailored electronic properties.

The 2D Young’s modulus of elasticity ($Y_{2D}$) and Poisson’s ratio ($v_{2D}$) have been determined using the following.
(2)$$Y_{2D}=\frac{C_{11}^{2}-C_{12}^{2}}{C_{11}},\;\;v_{2D}=\frac{C_{12}}{C_{11}}$$

Young’s modulus and Poisson ratio for the ScHBr_2_ (ScHI_2_) monolayer were found to be 27.77 (22.78) GPa and 0.26 (0.30), respectively. The Poisson ratio is less than 0.5 in both of the monolayers as the materials tend to become in-compressible for v2D>0.5. Also, the elastic constants for both the monolayers are in the range of the elastic constants reported for experimentally and theoretically predicted 2D monolayers (MoS_2_: 130 N/m (expt), WS_2_: 177 N/m (expt), Bi: 25.28 N/m (DFT), CdCl_2_: 30.0 N/m, (DFT)) [39].

To obtain the orientation angle-dependent Young’s modulus Y(θ) and Poisson’s ratio v(θ), we used the formulation [40,41]:(3)$$Y\left(\theta \right)=\frac{C_{11}C_{22}-C_{12}^{2}}{C_{11}\sin^{4}\left(\theta \right)+C_{22}\cos^{4}\left(\theta \right)+\left(C_{11}C_{22}-C_{12}^{2}\right)\cos^{2}\left(\theta \right)\sin^{2}\left(\theta \right)}$$
and
(4)$$v\left(\theta \right)=\frac{C_{12}\left(\sin^{4}\left(\theta \right)+\cos^{4}\left(\theta \right)\right)-\left(C_{11}+C_{22}-\frac{C_{11}C_{22}-C_{12}^{2}}{C_{66}}\right)\cos^{2}\left(\theta \right)\sin^{2}\left(\theta \right)}{C_{11}\sin^{4}\left(\theta \right)+C_{22}\cos^{4}\left(\theta \right)+\left(C_{11}C_{22}-C_{12}^{2}\right)\cos^{2}\left(\theta \right)\sin^{2}\left(\theta \right)}$$

From Figure 4, we can observe that, for the ScHBr_2_ monolayer, Young’s modulus, and Poisson ratio values are constant with varying θ, suggesting a negligible degree of anisotropy due to the absence of any deviation from the perfect circle. Thus, the monolayers exhibit mechanical isotropy, i.e., the Young’s modulus as well as the Poisson ratio, are orientation-independent. Similar variation has also been predicted in the ScHI_2_ monolayer in Figure A5.

### 3.4. Optical Properties

The optical properties of the monolayers can be obtained using the frequency-dependent dielectric function denoted as ε(ω), which represents the linear response of a system to the electromagnetic field. Its real ($\varepsilon_{1}\left(\omega \right)$) and imaginary ($\varepsilon_{2}\left(\omega \right)$) parts can be obtained using the Kramers-Kronig relations [42] and the absorption coefficient (α) is related to ε(ω) as follows:(5)$$\alpha \left(\omega \right)=\frac{\omega }{c}\sqrt{2\sqrt{\varepsilon_{1}^{2}\left(\omega \right)+\varepsilon_{2}^{2}\left(\omega \right)}-\varepsilon_{1}\left(\omega \right)}$$
where $\varepsilon_{1}\left(\omega \right)$ and $\varepsilon_{2}\left(\omega \right)$ are the real and imaginary parts of the frequency-dependent dielectric function, respectively.

Figure 5 shows the absorbance for the ScHX_2_ monolayers in the UV-VIS region. The ScHI_2_ monolayer shows a much higher absorption than the ScHBr_2_ monolayer. We would like to mention that the G0W0+BSE method provides a better description of the optical properties, as widely observed in the literature [43,44,45]. However, due to the computational intensiveness of this method, we were unable to employ it in the present study.

### 3.5. Strain Engineering

#### 3.5.1. Strain-Induced Modulation in Band Gap

Theoretical and experimental work has demonstrated that strain can play a significant role in manipulating the electronic, mechanical, and optical properties of bulk as well as 2D materials. However, traditional semiconductor bulk single crystals can only endure very limited strain, significantly constraining the application of strain modulation. In contrast, two-dimensional (2D) materials exhibit a much higher capacity for deformation and can withstand greater elastic strain without fracturing, making them highly promising candidates for strain engineering [46,47,48,49,50,51]. However, in experimental studies, the strain within the material may not be uniform, making it challenging to accurately evaluate atomic bond lengths and internal strain, especially on a statistical basis using large samples. To address this complexity, two primary methods are frequently employed to measure strain: local strain measurement and global strain measurement. Local strain measurement focuses on individual atomic bonds and their immediate environment, often using microscopic visual techniques such as transmission electron microscopy (TEM) or atomic force microscopy (AFM) to provide high-resolution images of the strained regions. On the other hand, global strain measurement evaluates the overall deformation of the material and often utilizes spectral measurement techniques like Raman spectroscopy or X-ray diffraction (XRD) to assess strain across a larger sample [47].

The strain-induced changes in bond length are directly correlated with changes in the energy within a material. When strain is applied, the atomic bonds can either stretch or compress, leading to variations in the total energy of the system (Figure A6). This relationship between bond length and energy is crucial for understanding how materials respond to mechanical deformation and for predicting their mechanical and electronic properties under different strain conditions.

On the application of the biaxial tensile strain, the bandgap reduces to 2.29 eV and 1.80 eV for ScHBr_2_ and ScHI_2_ respectively. However, with the application of the biaxial compressive strain, the band gap decreases from 2.49 to 1.37 eV for the ScHI_2_ monolayer and for the ScHBr_2_ monolayer, the band gap increases from 2.98 eV to 3.13 eV up to 5% biaxial compressive strain but then decreases as we go beyond 5% to 2.62 eV (Figure 6). To gain a deeper understanding of this change in trend, we have computed the orbital decomposed band structure at each strain value for the ScHBr_2_ monolayer, as illustrated in Figure 7. We can observe that with increasing the compressive strain, the lower valence bands shift towards the Fermi level (E = 0), leading VBM shift from the k point (Γ-M) to Γ. Moreover, we noticed a significant change in the band composition going from 5% to 6% biaxial compressive strain. At the 5% compressive strain, 61% contribution comes from the Br-*p* and 24% H-*s* orbitals at VBM. In contrast, at the 6% compressive strain, VBM shifts to Γ point with 80% contribution from the Br-*px* orbitals. Additionally, with the increase in the biaxial tensile strain, CBM composed of the Sc-*d* orbitals shifts towards the Fermi level, thereby lowering the band gap of the monolayer.

Also to confirm the stability of the strained monolayer we have also investigated the phonon bandstructure at 5% tensile as well as compressive strain (Figure A7). The absence of any significant negative phonon bands confirmed the dynamical stability of the strained monolayer.

For the ScHI_2_ monolayer, the variation in the orbital decomposed band structures has been displayed in Figure 8.

Application of the biaxial strain reduces the band gap to 1.76 eV for the (7%) compressive and 2.00 eV for the (7%) tensile strain (Figure 8). In the case of biaxial tensile strain, CBM shifts to the Γ (strain > 1%), with lowering the CBM close to the Fermi level which reduces the bandgap and VBM remains the same (Γ-M) (Figure 8). On the other hand, for the compressive strain, VBM shifts to Γ (strain > 1%), moving the CBM towards the Fermi level at the same k point.

To investigate the application of strain-engineering in the field of water-splitting we have compared the band edges to the oxidation and reduction potentials of water. Figure 9 depicts the alignment of VBM and CBM relative to the water redox potentials. A detailed discussion has been provided in the Appendix A. The results indicate the suitability of the energies of the band edges of the ScHBr_2_ monolayer for O_2_ and H_2_ evolution. However this is not the case for the ScHI_2_ monolayer, for which CBM does straddle the reduction potential but VBM doesn’t straddle the oxidation potential, as shown in Figure A8.

#### 3.5.2. Strain-Induced Modulations in Effective Masses

The effective mass of the charge carriers is a crucial parameter for devices as it plays a significant role in determining their transport properties and optical performance. It is determined by approximating the second derivatives of the VBM and CBM concerning the wave vector k and can be expressed as:(6)$$\frac{1}{m^{*}}=\frac{1}{\hbar^{2}}\left(\frac{d^{2}E}{dk^{2}}\right)$$

A flat band suggests the presence of heavy carriers or a high effective mass, whereas the well-dispersed bands indicate light carriers or a low effective mass. The strain affects the dispersion of the bands, reflected in a change in the effective masses of electrons and holes. Hence, the effective mass of electrons (holes) is determined by the dispersion of the energy bands at the CBM (VBM).

The effective masses for electrons and holes for both the monolayers for several strain values have been provided in Table 3. The effective mass is directly related to the carrier mobility, implying that the hole mobility in ScHX_2_ monolayers is predicted to be higher than that of electrons in the pristine ScHX_2_ monolayer.

In the case of the ScHBr_2_ monolayer, we can observe that the effective masses of the hole for the tensile as well as compressive strain remain almost similar since the curvature/dispersion of the bands remains nearly the same. However, a strain-induced range of variation is observed for the electron-effective masses. This variation is attributed to the change in the positions of the CBM as can also be observed from the orbital decomposed band structure in Figure 7. We can observe that for the tensile strain, the upper conduction band shifts towards the Fermi level. Hence the CBM now shifts to Γ for strain > 1%. It lowers the effective masses of electrons from 2.38 m_0_ to 0.83 m_0_ since now the conduction band has more dispersion at CBM. Whereas no shifting of the CBM is observed in the case of compressive strain hence the electron effective masses almost remain the same.

In contrast, for the case of the ScHI_2_ monolayer, Table 3 shows similar behavior for the biaxial tensile strain since the hole effective mass remains nearly the same. However, the hole becomes heavier with the tensile strain as the valance band becomes flatter at VBM (Figure 8). In contrast, for the case of the compressive strain, VBM shifts towards Γ leading to a decrease in the hole’s effective mass. A subsequent increase in the compressive strain leads to a decrease in the hole effective mass from 1.68 m_0_ to 1.09 m_0_ due to an increase in the dispersion of the valence band at VBM. Also, the application of the compressive strain makes the electron lighter hence the effective mass reduces to 0.77 m_0_ (7%). Whereas in the case of tensile strain, CBM remains almost flat at 3%, which increases the electron effective mass to 2.66 m_0_. A further increase in the tensile strain shifts CBM towards Γ, lowering the electron effective mass to 0.48 m_0_ (Table 3).

### 3.6. Substrate Supported ScHBr_2_ Monolayer

In general, the fabrication of an electronic device requires a substrate that is expected to be weakly bonded to the material. The choice of substrates is governed by lattice mismatch between the two, though the dry transfer method can be used for the large mismatched materials. In this study, our choice of substrate is a hexagonal 2D GaN monolayer as a substrate, which has been experimentally synthesized and theoretically studied [52,53,54]. Figure 10a displays the configuration of ScH${\mathrm{Br}}_{2}$/GaN heterostructure for which the lattice mismatch is calculated to be 11.6%. The inter-layer distance is 3.46 Å and the bandgap is 1.89 eV as shown in Figure 10b. The strained ScH${\mathrm{Br}}_{2}$ monolayer dominates the CBM and GaN denominates the VBM of the heterostructure hence forming a Type-II heterostructure.

The degree of interaction between the substrate and the monolayer can be quantified as follows:(7)$$E_{\mathrm{interaction}}=E_{\mathrm{heterostructure}}-E_{\mathrm{GaN}}-E_{\mathrm{monolayer}}$$
where $E_{\mathrm{heterostructure}}$, $E_{\mathrm{monolayer}}$, and $E_{\mathrm{GaN}}$ are the energies of the interface, monolayer, and the GaN substrate. The interaction energy was calculated to be −0.079 eV therefore indicating the feasibility of the formation of the heterostructure due to Equation (Equation 7) being exothermic. The interaction is mainly attributed to a charge transfer of 0.03 e from the monolayer to the substrate, as per Bader charge analysis. This is further confirmed by the charge density difference. The charge density difference calculated using Equation 8 also confirms a small charge transfer at the interface (Figure 11a)
(8)$$\rho_{r}=\rho_{\mathrm{interface}}-\left(\rho_{\mathrm{substrate}}-\rho_{\mathrm{monolayer}}\right)$$

The charge transfer is expected to induce a large intrinsic electric field across the interface which is displayed in Figure 11b with ΔU = 8.61 eV. Overall the substrate-induced modifications in the ScH${\mathrm{Br}}_{2}$ monolayer appear minimal retaining the semiconducting nature of the monolayer. Note that the calculated interaction energy of ScH${\mathrm{Br}}_{2}$ with the most commonly used graphene is predicted to be endothermic, thereby making the formation of the ScH${\mathrm{Br}}_{2}$/graphene heterostructure challenging.

## 4. Conclusions

In this study, novel mixed anion two-dimensional materials, ScHX_2_ (where X: Br/I), are investigated employing the density functional theory (DFT). Our findings indicate that the monolayers are stable according to various well-established parameters. It’s noteworthy that the electronic properties of these ScHX_2_ monolayers are comparable to well-established 2D materials like graphene and MoS_2_, making them highly suitable for electronic devices. Furthermore, we have comprehensively explored the mechanical, optical, and photocatalytic properties and have systematically compared them with the existing monolayers. Our investigation also delved into the influence of strain on charge carrier masses, offering insights into the potential mobility of charge carriers under varying strain conditions. The substrate-induced modification in the structural and electronic properties of the ScHBr_2_ monolayer appears minimal retaining the semiconducting nature of the monolayer.

## Figures and Tables

**Figure 1 nanomaterials-14-01390-f001:**
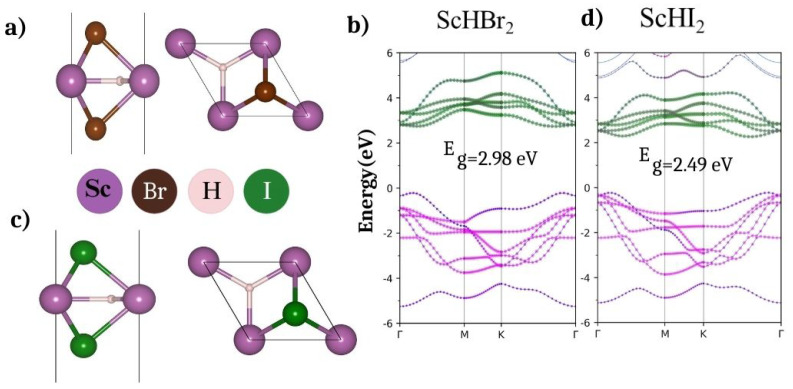
Side and top views of the unit cell and atom−projected band structure of (**a**,**b**) ScHBr_2_ (**c**,**d**) ScHI_2_ monolayers (atomic orbitals color code; Sc: green, H: blue and Br/I: pink).

**Figure 2 nanomaterials-14-01390-f002:**
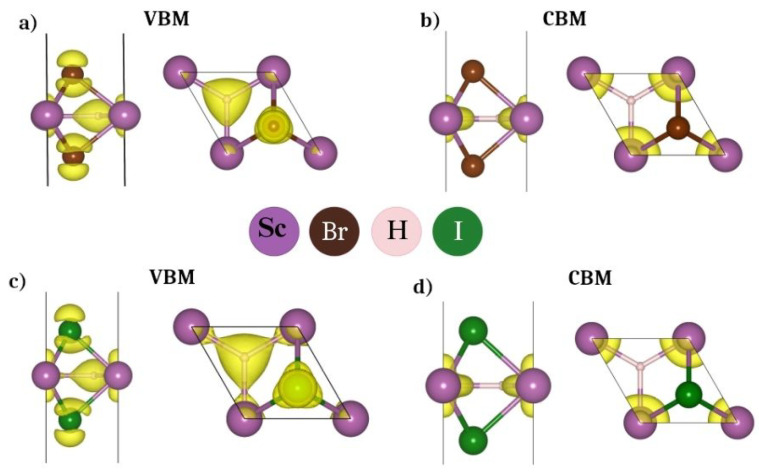
Top and side views of the band decomposed charge density at the valence band maxima (VBM) and conduction band minima (CBM) for (**a**,**b**) ScHBr_2_ and (**c**,**d**) ScHI_2_ monolayers at an isosurface value of $0.003\;{e/Å}^{3}$.

**Figure 3 nanomaterials-14-01390-f003:**
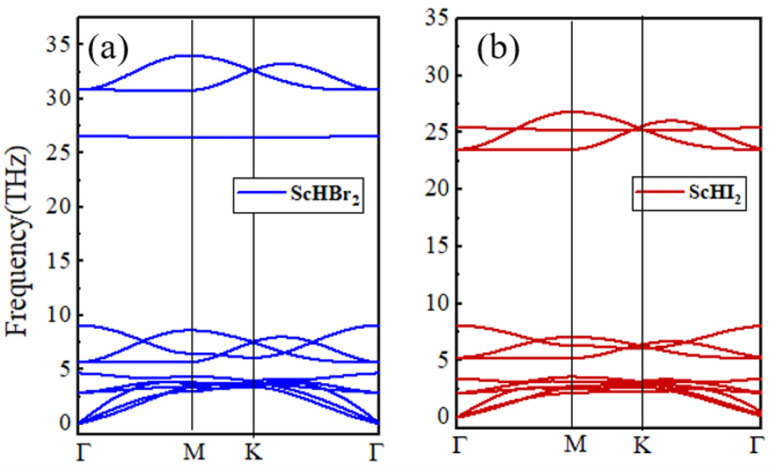
The phonon band structure of (**a**) ScHBr_2_ and (**b**) ScHI_2_ monolayer.

**Figure 4 nanomaterials-14-01390-f004:**
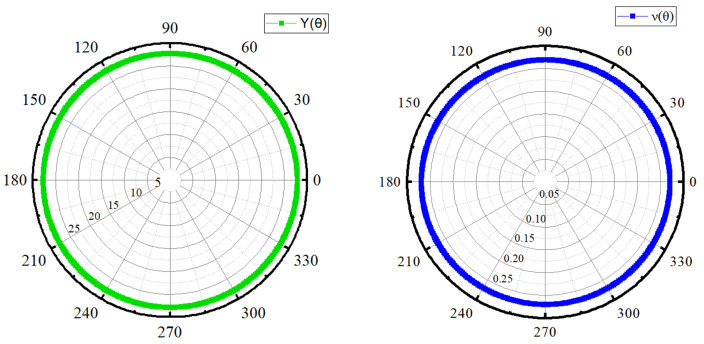
The calculated orientation dependence of Young’s modulus (GPa) Y(θ) and Poisson’s ratio v(θ) for the ScHBr_2_ monolayer.

**Figure 5 nanomaterials-14-01390-f005:**
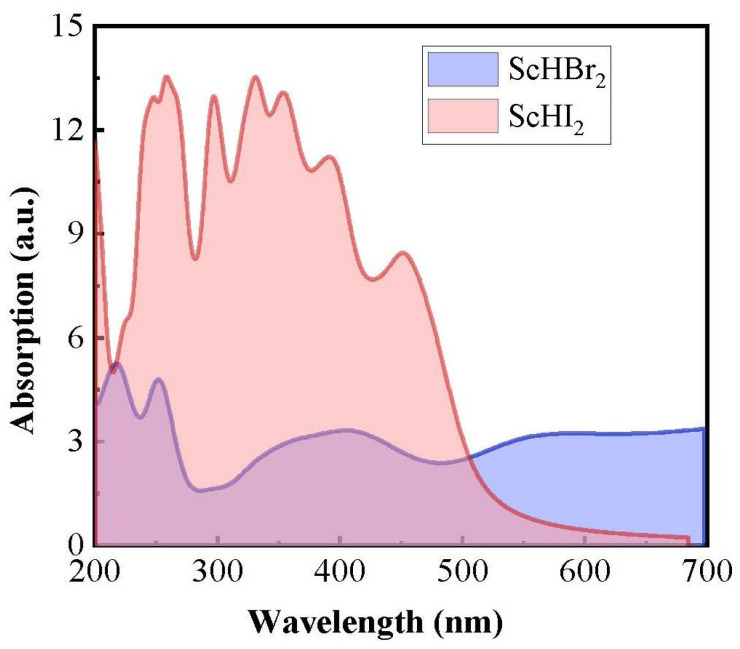
The absorption coefficient of ScHX_2_ monolayers in UV-VIS wavelength region.

**Figure 6 nanomaterials-14-01390-f006:**
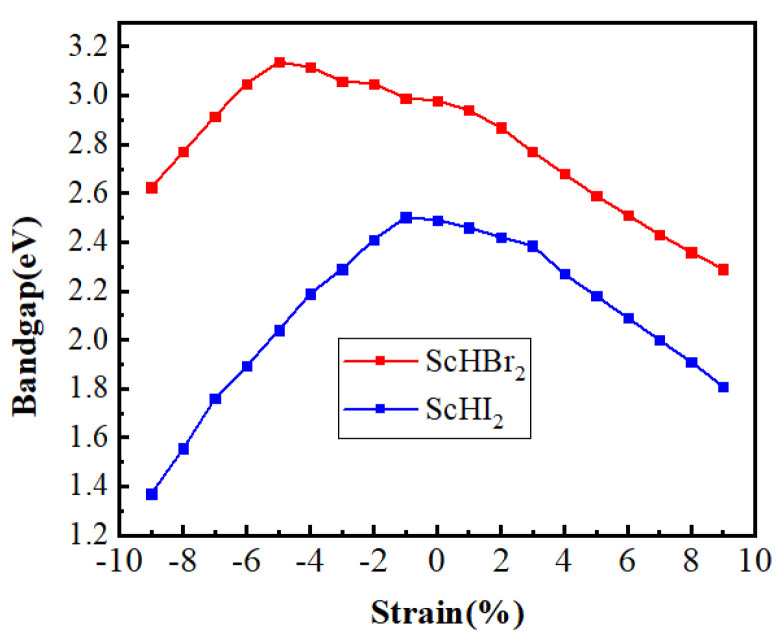
The calculated variation of the band gap in ScHX_2_ monolayers with tensile and compressive biaxial strain.

**Figure 7 nanomaterials-14-01390-f007:**
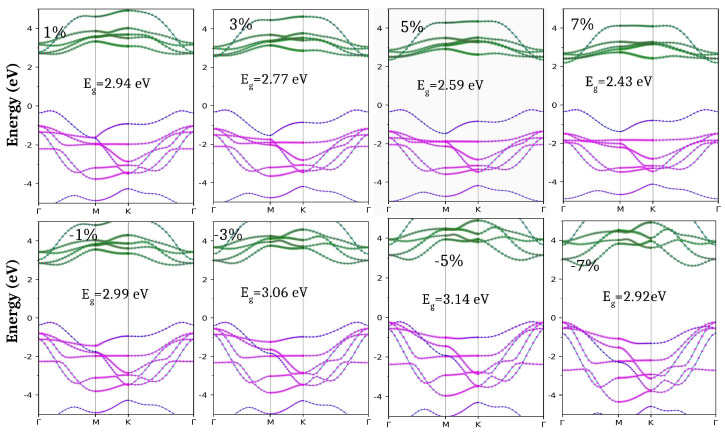
Atomic Orbital-projected band structure for ScHBr_2_ monolayer under the application of tensile and compressive biaxial strain using GGA-PBE. Color Code; Sc-*s*: yellow, Sc-*p*: brown, Sc-*d*: green, H-*s*: blue, I-*s*: cyan, I-*p*: pink and I-*d*: orange.

**Figure 8 nanomaterials-14-01390-f008:**
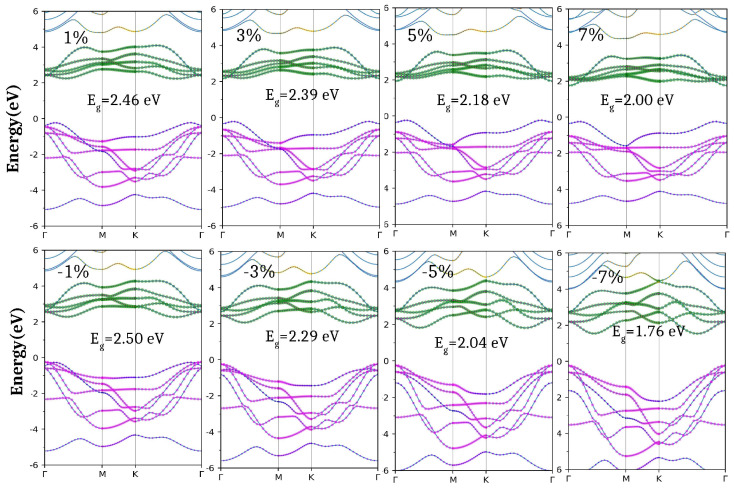
Atomic Orbital-projected band structure for ScHI_2_ monolayer under the application of tensile and compressive biaxial strain using GGA-PBE. Color Code; Sc-*s*: yellow, Sc-*p*: brown, Sc-*d*: green, H-*s*: blue, I-*s*: cyan, -*p*: pink and I-*d*: orange.

**Figure 9 nanomaterials-14-01390-f009:**
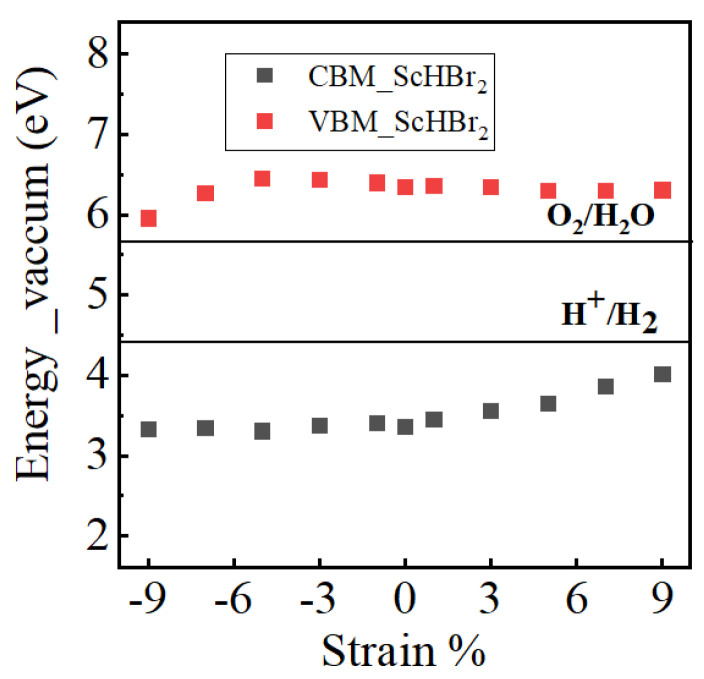
The calculated strain-induced variation in VBM and CBM positions with respect to the redox potential for the ScHBr_2_ monolayer.

**Figure 10 nanomaterials-14-01390-f010:**
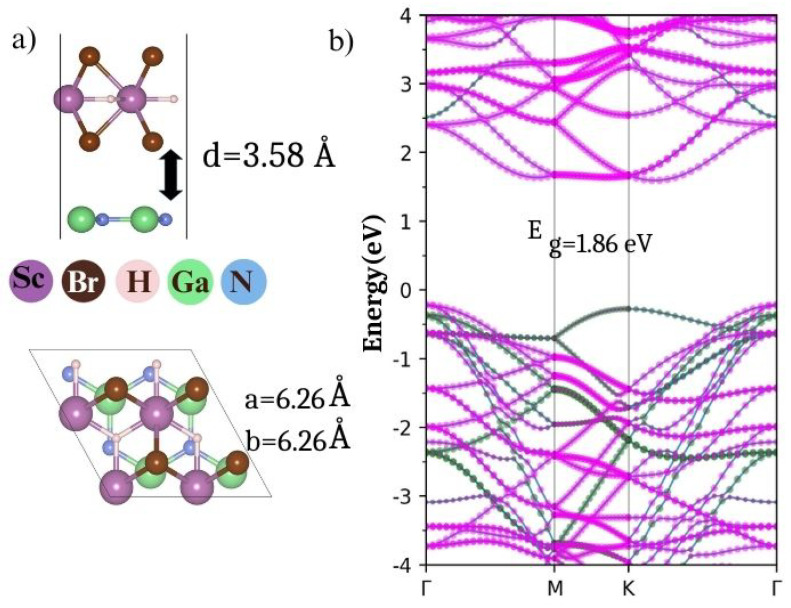
(**a**) Side and top views (**b**) band structure of the ScH${\mathrm{Br}}_{2}$/GaN heterostructure where the bands contributed by the ScH${\mathrm{Br}}_{2}$ monolayer are indicated by magenta color and bands contributed by GaN are indicated by emerald color.

**Figure 11 nanomaterials-14-01390-f011:**
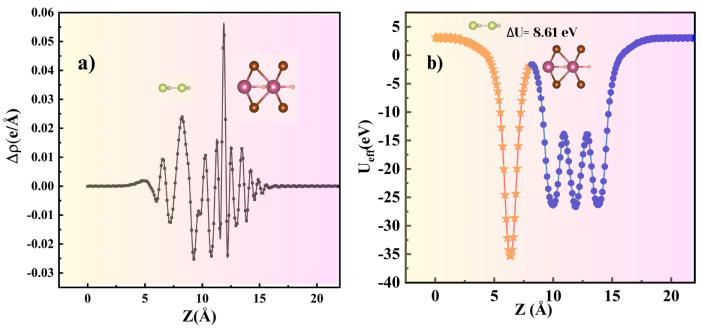
(**a**) Plane-averaged differential charge density (Δρ) (**b**) Plane-averaged electrostatic effective potential (U_eff_) along the z direction of the heterostructure.

**Table 1 nanomaterials-14-01390-t001:** Calculated Lattice Parameters (a = b), bond-lengths, Sc-H, Sc-Br/I, Ionisation Potential (IP), Work Function (ϕ), and Electron Affinity (EA) in ScH${\mathrm{Br}}_{2}$ and Sc${\mathrm{HI}}_{2}$ monolayers. * Refs. [24,25].

Monolayer	a = b	Sc-H	Sc-X	IP	ϕ	EA
	(Å)	(Å)	(Å)	(eV)	(eV)	(eV)
ScHBr_2_	3.62	2.08	2.78	6.35	5.99	3.37
				6.42 *^a^*	5.23 *^a^*	4.03 *^a^*
				6.09 *^b^*	4.61 *^c^*	3.62 *^b^*
ScHI_2_	3.82	2.20	2.99	5.62	5.40	3.13

*^a^*∗ MoS_2_, *^b^*∗ WS_2_, and *^c^*∗ Phosp.

**Table 2 nanomaterials-14-01390-t002:** The calculated elastic constants for the ScHX_2_ monolayers in GPa.

Monolayer	C_11_ = C_22_	C_12_	C_66_
ScHBr_2_	29.95	8.07	10.94
ScHI_2_	25.07	7.57	8.75

**Table 3 nanomaterials-14-01390-t003:** Electron and Hole Effective Masses for pristine and strained ScH$X_{2}$ monolayer.

Strain	%	ScHBr_2_		ScHI_2_	
		Electron	Hole	Electron	Hole
Compressive	7%	1.09	−1.05	0.77	−1.09
Compressive	5%	1.48	−0.63	0.91	−1.32
Compressive	3%	1.58	−0.65	1.11	−1.52
Compressive	1%	1.67	−0.77	1.45	−1.68
Pristine	0%	2.87	−0.98	1.57	−0.51
Tensile	1%	2.38	−0.65	1.94	−0.53
Tensile	3%	0.95	−0.72	2.66	−0.61
Tensile	5%	0.65	−0.76	0.44	−0.69
Tensile	7%	0.83	−0.86	0.48	−0.78

## Data Availability

The data is available in the Appendix A.

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
