# Peer review of "Strain and Substrate-Induced Electronic Properties of Novel Mixed Anion-Based 2D ScHX_2_ (X = I/Br) Semiconductors"

_nanomaterials, 2024, doi:10.3390/nano14171390_

Round 1

Reviewer 1 Report

Comments and Suggestions for Authors

The article "Strain and Substrate-induced electronic properties of novel Mixed Anion-based 2D ScHX2 (X=I/Br) Semiconductors" presents a DFT-based theoretical study on the structure and properties prediction of 2 novel materials - ScHBr2 and ScHI2 monolayers. Their stability based on several techniques is shown. Some basic properties such as band gap, optical spectra, mechanical characteristics etc are shown. The work can be suitable for publication if all comments below will be properly addressed.

1) There are serious drawbacks and shortcomings in writing the introduction part. Writing a comprehensive introduction can significantly enhance the readability of the work. At the first, the authors should present a comprehensive introduction about theoretical approaches and techniques
to predict 2D materials [please see https://doi.org/10.1088/2053-1583/ad2692].

2) Regarding computational technique, have the authors carried out energy cut-off convergence computations? Have they checked the Brillouin zone meshing convergence? Were the calculations spin-polarized and why?

3) Please extend the discussion on the simulation cell of ScHX2 monolayers. How were they created? If the structure is generated based on the bulk analogue?

4) Is the calculated electronic structure of monolayers in Figures 7 and 8 based on HSE functional? 

5) The authors should include the source crystallographic information (CIF) file of ScHX2 monolayers unit cells in the supplementary material file.

6) Why are all characteristics of 3 elemental ScHBr2 and ScHI2 monolayers compared to graphene or MoS2? There are many other 2D ternary materials, please extend the discussion of the result to make more reasonable comparisons. For example, see https://doi.org/10.1038/s41524-020-00428-x.

7) The author should discuss the potential impact on their results.

Author Response

Comment 1.1 There are serious drawbacks and shortcomings in writing the introduction part. Writing a comprehensive introduction can significantly enhance the readability of the work. At first, the authors should present a comprehensive introduction about theoretical approaches and techniques to predict 2D materials [please see https://doi.org/10.1088/2053-1583/ad2692].

Response 1.1 We would like to thank the reviewer for the comments. We have revised the introduction (Page 2) to include the theoretical approach of high throughput computing in predicting 2D materials.

Comment 1.2 Regarding computational technique, have the authors carried out energy cut-off convergence computations? Have they checked the Brillouin zone meshing convergence? Were the calculations spin-polarized and why?

Response 1.2. The convergence criterion for the energy cut-off (ENCUT) and K-mesh values have been checked as shown below:

Fig 1. Total Energy variation with cut-off energy (ENCUT) and K-Mesh of ScHBr2 monolayer.

Comment 1.3 Please extend the discussion on the simulation cell of ScHX2 monolayers. How were they created? If the structure is generated based on the bulk analogue?

Response 1.3.

Page 3: The details of the simulation cell and the structural properties are given in Result  Section and Table 1. The 2D structure for ScHX2 was created using the bulk analog (LaHBr2), which has been experimentally synthesized (Zeitschrift für anorganische und allgemeine Chemie 1992, 607, 29–33) and also studied using high throughput screening. (arXiv:2307.03933 2023). Accordingly, the lattice comprises 1 Sc, 1H, and 2 X(Br/I) atoms, and the simulation box consists of a hexagonal unit cell with lattice constants a=b and c > 20 Å, and the α=β=90°, γ=120°.

Comment 1.4 Is the calculated electronic structure of monolayers in Figures 7 and 8 based on the HSE functional? 

Response 1.4.  No. We used the GGA-PBE functional form to study variation in the band structure under strain. Our aim was to explore how strain impacts the bandgap and to assess whether the material retains its semiconducting nature.

We have revised the captions of Figures 7 and 8.

Comment 1.5 The authors should include the source crystallographic information (CIF) file of ScHX2 monolayers unit cells in the supplementary material file.

Response 1.5. Agreed.  We have included the VASP structure file for ScHX2 monolayers at the end of the SI.

Comment 1.6 Why are all characteristics of 3 elemental ScHBr2 and ScHI2 monolayers compared to graphene or MoS2? There are many other 2D ternary materials, please extend the discussion of the result to make more reasonable comparisons. For example, see https://doi.org/10.1038/s41524-020-00428-x.

Response 1.6. Thank you for the comment.

Page 3: A comparison of the bandgap with other 2D materials is now discussed.

Comment 1.7 The author should discuss the potential impact on their results.

Response 1.7 The potential impact has been discussed in the Conclusion Section (Page12)

The results predict that ScHX2 monolayers are highly suitable for electronic devices, and their properties are comparable to well-established 2D materials like graphene and MoS2; we find them to be stable and investigated the influence of strain on charge carrier masses to gain an insight into the mobility of charge carriers under varying strain conditions. Moreover, the substrate-induced modifications in their properties appear minimal, suggesting that the fabrication of ScHX2-based devices will retain their semiconducting nature.

Reviewer 2 Report

Comments and Suggestions for Authors

In this paper, ScHX2 (where X: Br/I) are investigated by employing the density functional theory (DFT). The electronic and mechanical properties are comparable to graphene and MoS2. The computational results may be interesting to the relative researchers. I think this study is publishable but there are issues that must be addressed in the revised submission.

1.          In terms of parameter selection for calculations, the paper should discuss the impact of key parameters, such as energy cutoff and k-point grid density, on the electronic structure and total energy.

2.          The lattice mismatch of ScHBr2/GaN heterostructure is calculated to be 11.6%, will it affect the crystal quality and electronic structure of the films?

3.          In the comprehensive analysis of strain effects, strain engineering provides an effective means to regulate the electronic and optical properties of ScHX2 materials. However, further research is needed on the stability of ScHX2 monolayer materials.

Comments on the Quality of English Language

Minor editing of English language required

Author Response

Comment 2.1 In terms of parameter selection for calculations, the paper should discuss the impact of key parameters, such as energy cutoff and k-point grid density, on the electronic structure and total energy.

Response 2.1. Thank you for your comment. The convergence criterion for the energy cut-off (ENCUT) and K-mesh values has been checked, as shown in Figure 1 (Response 1.2).

Comment 2.2 The lattice mismatch of ScHBr2/GaN heterostructure is calculated to be 11.6%, will it affect the crystal quality and electronic structure of the films?

Response 2.2.  The strained ScHBr2 monolayer will retain the semiconducting nature with a bandgap of 1.86 eV.

Comment 2.3 In the comprehensive analysis of strain effects, strain engineering provides an effective means to regulate the electronic and optical properties of ScHX2 materials. However, further research is needed on the stability of ScHX2 monolayer materials.

Response 2.3. We have evaluated the stability of the ScHX₂ monolayers using multiple criteria. In addition, the stability of the strained (5% biaxial compressive and tensile strained) ScHBr2 monolayer has also been confirmed. (Fig. S7, SI)

Round 2

Reviewer 1 Report

Comments and Suggestions for Authors

The authors fully addressed all my comments and significantly improved the manuscript. I recommend the manuscript for publication.